# Molecular Mechanisms Determining the Role of Bacteria from the Genus *Azospirillum* in Plant Adaptation to Damaging Environmental Factors

**DOI:** 10.3390/ijms24119122

**Published:** 2023-05-23

**Authors:** Maria V. Gureeva, Artem P. Gureev

**Affiliations:** 1Department of Biochemistry and Cell Physiology, Voronezh State University, 394018 Voronezh, Russia; maryorl@mail.ru; 2Department of Genetics, Cytology and Bioengineering, Voronezh State University, 394018 Voronezh, Russia; 3Laboratory of Metagenomics and Food Biotechnology, Voronezh State University of Engineering Technology, 394036 Voronezh, Russia

**Keywords:** *Azospirillum*, stress, hydrocarbons, heavy metals, phytopathogens, pesticides, perchlorate, drought, salinization

## Abstract

Agricultural plants are continuously exposed to environmental stressors, which can lead to a significant reduction in yield and even the death of plants. One of the ways to mitigate stress impacts is the inoculation of plant growth-promoting rhizobacteria (PGPR), including bacteria from the genus *Azospirillum*, into the rhizosphere of plants. Different representatives of this genus have different sensitivities or resistances to osmotic stress, pesticides, heavy metals, hydrocarbons, and perchlorate and also have the ability to mitigate the consequences of such stresses for plants. Bacteria from the genus *Azospirillum* contribute to the bioremediation of polluted soils and induce systemic resistance and have a positive effect on plants under stress by synthesizing siderophores and polysaccharides and modulating the levels of phytohormones, osmolytes, and volatile organic compounds in plants, as well as altering the efficiency of photosynthesis and the antioxidant defense system. In this review, we focus on molecular genetic features that provide bacterial resistance to various stress factors as well as on *Azospirillum*-related pathways for increasing plant resistance to unfavorable anthropogenic and natural factors.

## 1. Introduction

Farming is a risky industry, the result of which largely depends on environmental factors. Agricultural plants are regularly exposed to a variety of stress factors. This may be drought, infection by pathogenic micro-organisms, growth on saline soils or on soils contaminated with hydrocarbons, heavy metals, pesticides, radioactive elements, or perchlorate. One of the ways to reduce the stress impact on plants and increase their productivity is the use of plant growth-promoting rhizobacteria (PGPR) [1]. PGPR are the rhizosphere bacteria that can enhance plant growth via a wide variety of mechanisms, such as phosphate solubilization, siderophore production, biological nitrogen fixation, rhizosphere engineering, the production of 1-Aminocyclopropane-1-carboxylate deaminase (ACC), the quorum sensing signal interference, the inhibition of biofilm formation, phytohormone production, exhibiting antifungal activity, the production of volatile organic compounds, the induction of systemic resistance, promoting beneficial plant–microbe symbioses, interference with pathogen toxin production, etc. [2,3]. 

Depending on the degree of association of bacteria with plant root cells, PGPR can be divided into extracellular plant growth-promoting rhizobacteria (ePGPR) and intracellular plant growth-promoting rhizobacteria (iPGPR) [4]. ePGPR includes such genera as *Agrobacterium*, *Arthrobacter*, *Azotobacter*, *Azospirillum*, *Bacillus*, *Burkholderia*, *Caulobacter*, *Chromobacterium*, *Erwinia*, *Flavobacterium*, *Micrococcus*, *Pseudomonas*, and *Serratia* [5]. iPGPR includes endophytes (*Allorhizobium*, *Azorhizobium*, *Bradyrhizobium*, *Mesorhizobium*, and *Rhizobium*) and *Frankia* species [6].

Microbial inoculants consisting of PGPR are the most widely used in Latin America, Southeast Asia, and Africa, where inoculated seeds are sown on a large scale, with millions of hectares of *Fabaceae* (e.g., soybean or bean) and *Poaceae* (e.g., maize, sorghum, or wheat) inoculated through PGPR, belonging mainly to the genera *Bacillus*, *Paenibacillus*, *Pseudomonas*, or *Azospirillum* [7]. Each of the most commonly used genera is most effective in some way for plant growth promotion: phytohormone production (e.g., *Azospirillum* spp., *Pseudomonas* spp.), phosphate dissolution (e.g., *Bacillus* spp.), or biological control (e.g., *Pseudomonas* spp., *Bacillus* spp.) [8]. 

A number of reviews on the use of bacteria from the genera *Bacillus* [9,10,11,12,13,14,15,16], *Paenibacillus* [17,18], and *Pseudomonas* [19,20,21] for plant growth promotion have been published in the last decade. In particular, the role of these bacteria in the protection against biotic and abiotic stresses [9,12,19] and molecular mechanisms that determine their interactions with plants [14,15,21] were considered. 

As for bacteria from the genus *Azospirillum*, the last review on the molecular basis of the interaction of the representatives of this genus with plants was published in 2012 [22]. In 2019, a review was published on the use of *Azospirillum* as inoculants in crop plants [8], but it focused more on the effectiveness of using commercial inoculants in different countries rather than the molecular mechanisms of their effect on plants. In 2018, a review was published on the role of azospirilla in the protection against biotic stress as well as two types of abiotic stress, namely osmotic and oxidative [23]. In this review, we describe the role of azospirilla in mitigating stress in plants caused by hydrocarbons, perchlorate, radiation, heavy metals, and pesticides and add information about the latest data on the mechanisms of biotic and osmotic stress mitigation. 

## 2. Bacteria from the Genus *Azospirillum*

The genus *Azospirillum* currently includes 28 species, 24 of which have validly published names (https://lpsn.dsmz.de/genus/azospirillum accessed on 25 April 2023). Most azospirilla species were isolated from the soil or plant rhizosphere, although individual species were isolated from water bodies, oil-producing mixtures, discarded tar, fermented cattle products, fermenter, microbial fuel cells, and karst caves.

*Azospirillum* is one of the most studied genera of PGPR, and species of this genus are recognized as biofertilizers due to their ability to stimulate plant growth and productivity [24]. Bacteria of this genus are resistant to many types of biotic and abiotic stress and are also capable of activating plant defense mechanisms upon inoculation into the rhizosphere, increasing crop yields in stress conditions (Table 1).

### Azospirillum Response to Stress

The stress response in many bacteria is activated via the extracytoplasmic function σ factors (ECF). Due to their diversity and relative simplicity of the mechanism of action, they stand out as a versatile and powerful bacterial tool for the effective activation of stress responses [45]. They are subunits of the RNA polymerase holoenzyme required for transcription initiation. ECFs belong to group IV σ factors and consist of two domains, σ2 at the N-terminus and σ4 at the C-terminus. Upon transcription initiation, σ2 binds to the −10-box in the promoter, while σ4 binds to the −35-box, and two-stranded DNA begins to melt in the −10-box. ECF activity can be regulated through anti-σ factors, through serine/threonine protein kinases, and through C- and N-terminal extensions. In addition, some ECFs may be regulated only by controlling their production at the transcriptional level [45].

The regulation of the stress response in the representatives of the genus *Azospirillum* was studied using the strains of the species *A. brasilense*. It was shown that adaptation to many types of stress is mediated through ECF, which can be regulated by anti-sigma factors (Figure 1). The role of ECF, known as RpoE or σ E, in the adaptation to salt, ethanol, and methylene blue stress was shown for *A. brasilense* Sp7 [46].

The synthesis of carotenoids in response to stress in *A. brasilense* is regulated by ECF rpoE, which, in turn, is regulated by the anti-sigma factor chrR [47,48]. Kumar et al., 2012, showed that ECF RpoH2 controls the response to photooxidative stress in *A. brasilense* [49]. Gupta et al., 2014, showed that *A. brasilense* contains two redox-sensitive zinc-binding anti-sigma factors (ZAS) (ChrR1 and ChrR2), which negatively regulate the activity of their related ECFs (RpoE1 and RpoE2), blocking their binding to bovine enzyme. At the same time, two *A. brasilense* ZAS anti-σ factors also interact with their unrelated ECFs and affect gene expression [50].

ECF RpoH2 in *A. brasilense* regulates the use of ethanol as an additional carbon source when growing on fructose or glycerol [51]. Pandey et al., 2022 described a new ECF RpoE7-RpoH3 regulatory cascade that negatively regulates ampicillin resistance in *A. baldaniorum* Sp245 by controlling the expression of β-lactamase and lytic transglycosylase [52].

Authors also paid attention to ECF-encoding genes in works on the sequencing and analysis of the genomes of azospirilla. The genome of *A. brasilense* Sp7 encodes one home and twenty-two alternative ECFs, consisting of ten RpoE, five RpoH, one RpoN, and six FecI sigma factors [53]. Fourteen *rpoE* genes and five *rpoH* genes were found in the genome of *A. brasilense* Az19 [54].

## 3. *Azospirillum* Participation in Plants’ Defense against Stress Factors

### 3.1. Hydrocarbon Pollution

Hydrocarbons are the largest group of organic pollutants. The increasing dependence of humanity on fossil fuels, especially petroleum hydrocarbons, has led to the pollution of agricultural lands through the spillage of crude oil during extraction and processing operations in many oil producing countries [55]. These hydrocarbons are highly resistant, can accumulate in plants, as well as in humans and animals, and exhibit carcinogenic and neurotoxic properties [56]. One of the ways to effectively remove hydrocarbons from the soil is microbial biodegradation.

Bacteria from the genus *Azospirillum* are found in microbial communities that break down hydrocarbons [57]. There are few data on the ability of individual strains to remove oil. Some *Azospirillum* strains have been shown to biodegrade crude oil [58], phenol, and benzoate [59], as well as polycyclic organic compounds [57,60,61,62,63,64]. Additionally, representatives of the genus *Azospirillum* were found in biofilms that decompose hydrocarbons [65,66] and, as part of the microbiome in the maize rhizosphere, bioremediate soil contaminated with crude oil [67]. It has been suggested that this bacterium appears to enrich biofilms with nitrogenous compounds known to enhance the microbiological degradation of hydrocarbons [66]. 

In addition, two *Azospirillum* species isolated from oil-bearing samples, *A. rugosum* [68] and *A. oleiclasticum* [24], were described. For the latter species, the ability to biodegrade crude oil was shown [24].

Thus, the metabolic potential of the genus *Azospirillum* allows its representatives to participate in the biodegradation of hydrocarbons, thereby contributing to the bioremediation of polluted soil and, consequently, reducing the damaging effect of this pollutant on plants. 

### 3.2. Heavy Metal Pollution

Heavy metals are an essential part of the environment, but in places of active anthropogenic activity, their concentration significantly exceeds the permissible limits, which adversely affects agriculture [69]. In plants, heavy metal stress has both direct and indirect effects, including oxidative stress through various indirect mechanisms (e.g., the depletion of glutathione or its binding to protein sulfhydryl groups) or through the inhibition of antioxidant enzymes, thereby inducing ROS (reactive oxygen species)-producing enzymes (for example, NADPH oxidases) [70].

PGPR biostimulants are incredibly effective at reducing heavy metal toxicity in plants. They inhibit the transfer of heavy metals to various areas of the plant, changing their mobilization through complexation, precipitation, redox processes, chelation, and adsorption [71,72,73]. In addition, rhizospheric bacteria produce extracellular polymeric substances (EPS) [74,75], such as polysaccharides, glycoproteins, lipopolysaccharides, and soluble peptides, which contain many anion binding sites, and thus contribute to the displacement or recovery of heavy metals from the rhizosphere through biosorption [76].

Bacteria from the genus *Azospirillum* are able to tolerate high concentrations of heavy metals: arsenic [77,78,79], cadmium [26,27,80], copper [81], and lead [80]. Moreover, bacteria can reduce the negative effects of heavy metals on plants growing in contaminated soil (Figure 2). Vezza et al., 2019, showed that arsenic-resistant genes can mediate the redox transformation of As and its displacement outside the cell [77].

Peralta et al., 2021, showed different effects of different strains of *A. brasilense* on the content of photosynthetic pigments in maize in the presence of arsenic: strain CD caused their significant decrease, while strain Az39 did not affect their amount [82]. The use of *A. brasilense* as biological additives reversed the effects of arsenic toxicity by increasing wheat plant growth rate, leaf area, and photosynthesis, and yield [25]. Additionally, the co-inoculation of soybean seeds with the bacteria *Bradyrhizobium japonicum* E109 and *A. brasilense* Cd had a positive effect on nodule formation, photosynthetic pigment content, and antioxidant system activity, as well as a significant reduction in the accumulation of arsenic in plant tissues exposed to AsV and AsIII [77]. 

It has been shown that bacteria from the genus *Azospirillum*, alone or in combination with another rhizosphere bacterium, *Bacillus subtilis*, are able to reduce cadmium toxicity for arabidopsis, pakchoi, and barley [26,27,28,29]. A decrease in the concentration of cadmium in plants and an increase in the biomass of shoots occurred due to an increase in the concentration of abscisic acid (ABA) [26,27,29]. The action of ABA was mediated through IRT1 (IRON-REGULATED TRANSPORTER 1) [26]. A decrease in the level of cadmium toxicity for plants could also be due to a decrease in oxidative stress and an increase in the activity of antioxidant enzymes [29].

Bacteria from the species *A. brasilense* are able to reduce copper stress in wheat [81], cucumber [30], and an algae *Chlorella sorokiniana* [83] by activating antioxidant defense enzymes. Moreover, it has been shown for wheat that the copper content in plants increases upon inoculation with azospirilla but its toxicity decreases [81]. For *Chlorella sorokiniana*, it has also been shown that inoculation with azospirilla increases the content of chlorophyll due to the secretion of IAA (indoleacetic acid) [83]. The ability of bacteria from the genus *Azospirillum* to produce auxin affects the accumulation of zinc and iron in corn in different ways: a low ability of azospirilla to produce auxin leads to an increase in the zinc content in plants and a high ability leads to an increase in the iron content [84].

Thus, several main mechanisms of reducing the toxicity of heavy metals to plants by bacteria from the genus *Azospirillum* can be identified: through a decrease in oxidative stress, through an increase in the activity of antioxidant enzymes and the amount of photosynthetic pigments, and through the regulation of the amount of phytohormones.

### 3.3. Infection of Plants with Phytopathogens

Plant pathogens have a negative impact on the marketable yield (i.e., quality and quantity) of agricultural products, with an adverse impact on the economy. Approximately 14% of crops worldwide are killed by disease, and worldwide crop losses can be as high as 20–40% in sensitive strains [85]. In this regard, the issue of protecting agricultural plants from pathogens is of great importance.

Traditional methods of controlling plant pathogens include implementing good agricultural practices that prevent further infestation; the physical destruction of infected plant tissues; the use of chemicals, such as pesticides and antibiotics, to fight bacterial infections; the development of genetically modified plants resistant to pests and pathogens; and the use of bacteriophages [85].

In addition to the above methods, the PGPR inoculation of agricultural plants has been actively used in recent years to reduce the negative effect of phytopathogens. In particular, the bacteria of the genus *Azospirillum* have been shown to be capable of the biological control of phytopathogens [31,32,33,34,86]. This may be due to the synthesis of siderophores that limit the availability of iron (Fe) to phytopathogens [86] or the induction of changes in the host plant metabolism, which increases plant resistance to pathogen infection—the induced systemic resistance (ISR) [23].

Siderophores are compounds with low molecular weight (<1500 Da) and high iron affinity that allow soil micro-organisms to bind and dissolve ferric iron in iron-poor environments. The conversion of iron into an available form and the subsequent increase in the uptake of the available form of iron by plants can lead to the prevention of the growth of soil pathogens due to iron deficiency. Siderophores vary greatly in chemical structure; however, they can be divided into two main groups, namely catechols and hydroxamates, according to the chemical group involved in iron(III) chelation [86].

Among the catechols, salicylic acid (SA) has received particular attention, because it can be active in pathogen biocontrol in two ways. On the one hand, it can act as a siderophore, reducing the availability of iron in an environment with a low iron content [87], and on the other hand, it can act as a signal molecule that triggers a systemic response of plant resistance to pathogens [88]. It is the synthesis of catechol siderophores, including SA, that allows *A. brasilense* to exhibit antifungal activity against *Colletotrichum acutatum*, the causative agent of anthracnose, and reduce its negative effect on strawberry plants [86]. Additionally, the synthesis of siderophores via the bacterial strains of *A. brasilense* is able to determine the resistance of the teosinte plant (*Zea mays* L. ssp. *mexicana*) to the phytopathogenic fungi *Alternaria* (causative agent of Alternaria), *Bipolaris* (causative agent of helminthosporiasis), and *Fusarium* (causative agent of Fusarium) [35].

Another form of *Azospirillum* limitation regarding the development of phytopathogens is the induction of systemic resistance in plants. Plant systemic resistance can be divided into ISR and systemic acquired resistance (SAR) induced by non-pathogenic microbes and pathogenic microbes, respectively [89,90]. Colonization with beneficial microbes induces a physiological state of the host plant called “priming”. When “priming” is activated, plants exhibit stronger and faster defense responses against the subsequent pathogen invasion [91].

The classic difference between ISR and SAR, adopted in 1996, is the type of activated signaling pathway. For ISR, these are the jasmonic acid (JA) and ethylene (ET) pathways, and for SAR, these are the SA pathway and the activation of PR (pathogenesis-related) proteins [92]. However, there have been numerous reports of the activation of both the SA and JA/ET signaling pathways in ISR triggered by beneficial microbes [91]. As for PR proteins, the activation of PR1, PR2, and PR5 depends on SA signaling, while PDF1.2, as well as the PR3 and PR4 genes, are activated via an SA-independent and JA-dependent pathway [93].

In the SA pathway, the activation/repression of PR genes is mediated by NPR1 (“nonexpressor of PR-gene1”, related to the plant’s defense system). When SA levels are low, NPR4 (paralog of NPR1) interacts with NPR1, resulting in its degradation. Thus, when SA levels are high, binding between NPR1 and NPR3 (paralog of NPR1) is increased, which also leads to the removal of NPR1 [94]. When SA is intermediate, the interaction between NPR1 and NPR3 is suppressed, resulting in the accumulation of NPR1 and the activation of SA-dependent protective genes [95].

The major players In the JA pathway are the CORONATINE INSENSITIVE 1 (COI1) protein, JASMONATE ZIM DOMAIN PROTEIN (JAZ), and MYC. In the absence of stress, the endogenous level of the active form of JA, isoleucine jasmonate (JA-Ile), is very low in plants. JAZ repressors bind to MYC2 to inhibit its transcriptional activation on downstream genes. Under stress conditions, the endogenous level of JA-Ile is activated to a large extent, which is perceived by the JA-receptor COI1. SKP1/CULLIN/F-box (SCF)COI1 then binds to JAZ for ubiquitination and degradation via the 26S proteasome pathway, resulting in the release of downstream transcription factors, such as MYC, and the activation of JA responses [96].

The classical ET pathway is a linear sequence of the following components: the ET receptor family; the protein kinase CTR1; the transmembrane protein with unknown biochemical activity, EIN2; the transcription factors EIN3, EIL and ERF; and the ET response. In the absence of ET, the receptors activate CTR1, which negatively regulates downstream signaling [97].

There have been several attempts to identify the signaling pathways leading to the emergence of systemic plant resistance upon inoculation with bacteria from the genus *Azospirillum*. In a study of strawberries (*Fragaria ananassa*) inoculated with *A. brasilense* REC3, Elias et al. (2018) reported increased ET synthesis and the upregulation of genes associated with ET signaling (*Faetr1*, *Faers1*, *Faein4*, *Factr1*, *Faein2*, and *Faaco1*) [98]. Kusajima et al., 2018, also showed that *A. brasilense* induces ISR in rice through the ET pathway [99]. Yasuda et al. (2009) showed that rice plants inoculated with *Azospirillum* sp. B510 increased resistance to the pathogenic fungus *Magnoporthe oryzae* (the causative agent of blast) and to the bacteria *Xanthomonas oryzae* (the causative agent of bacterial blight of rice) through the mechanisms independent of SA signaling, without the accumulation of SA or PR proteins [100].

However, other studies showed that PR proteins play a role in the formation of systemic resistance in plants in response to inoculation with azospirilla. A transcriptome study showed that *Azospirillum* sp. strain B510 (isolated from cv. Nipponbare) inoculated into rice induced one and repressed five PR genes, while strain *A. lipoferum* 4B (isolated from cv. Cigalon) induced more protection-related genes in rice cv. Nipponbare than in rice cv. Cigalon [101]. In another study with *Arabidopsis thaliana*, PR genes were induced when the plant was inoculated with *A. brasilense* Sp245 [102]. A study was also conducted using *A. brasilense* Ab-V5 and Ab-V6 cells and metabolites, which led to the induction of PR-1 SAR-associated genes and PRP-4 ISR-associated genes [103].

Thus, at present, it is not possible to draw an unambiguous conclusion about the systemic resistance pathway induced by azospirilla in inoculated plants. Most likely, this is a combination of different pathways, and their relationship and regulation needs to be studied in more detail.

### 3.4. Pesticide Pollution

The third agricultural revolution, or green revolution, which took place in the second half of the 20th century, made it possible to significantly increase the productivity of many agricultural crops. Much of this was made possible through the widespread use of pesticides [104]. However, only 1% of the pesticide reaches the pest, while the rest accumulates in soil, water, and air and affects non-target organisms, including agricultural plants [105]. Pesticides accumulate in the plant body and can target the electron transport chains in photosystems in chloroplasts [106], inhibit respiratory complexes in mitochondria, uncouple phosphorylated respiration, damage DNA [107], cause oxidative stress [108], disrupt the metabolism of polyphenols, reduce the bioavailability of trace elements [109], and negatively influence rhizospheric bacteria [110].

Data on pesticide toxicity for azospirilla are inconsistent and not abundant. Several works on this subject were carried out at the end of the 20th century and beginning of the 21st century. In vitro studies showed that methidathion is able to reduce nitrogen fixation, intracellular ATP levels, and cell growth, while profenophos also inhibits the production of a number of hormones in *A. brasilense* [111]. At the same time, terbufos has little effect on the growth of *A. lipoferum* on a solid medium, while carbofuran, chlormephos, and benfuracarb do not affect it at all [112]. Bromopropylate and diazinon are also completely harmless to *A. brasilense* [111].

Under field conditions, the population of *Azospirillum* sp. decreased in vigna treated with thiram but not in plants treated with carbendazim, Bordeaux mixture, carbofuran, and phorate. A mixture of thiram and carbofuran and phorate reduced the population of azospirilla, but after treatment, a gradual accumulation of bacteria was observed in the rhizosphere [113]. Additionally, the soil isolates of *Azospirillum* sp. were able to degrade the pesticide Ethion [114].

In recent years, there has been renewed interest in research on the interaction of azospirilla and pesticides regarding the joint treatment of cereal seeds before sowing. The treatment of plant seeds with pesticides Standak™ Top (BASF) (a mixture of insecticide fipronil and fungicide pyraclostrobin and thiophanate-methyl) and Helicur 250 EW (tebuconazole) is known to reduce the survival of *Azospirillum* bacteria [115,116]. It has been shown in terms of insecticides (imidocloprid and thiodicarb) and fungicides (triadimenol) that azospirilla can survive only if the interval between the inoculation of pesticide-treated seeds and sowing in the soil does not exceed 4 h [117]. 

Thus, the joint treatment of seeds with azospirilla and pesticides is possible; however, for each pesticide, it is necessary to choose compatible strains and it is necessary to follow a certain treatment technology that preserves the viability of the strains used.

### 3.5. Pollution with Radioactive Elements

There was an attempt to inoculate plants with *Azospirillum* strains in contaminated soil in Fukushima for the purpose of bioremediation by translocating radioactive caesium to the aerial parts of the plants. Despite the positive effects of inoculation, the concentrations of (137)Cs during their transfer to the tested plants were not very high, and the removal of (137)Cs from the soil would therefore be very slow [36].

### 3.6. Perchlorate Pollution

Perchlorate is a persistent pollutant produced by natural and human processes [118]. Perchlorates were shown to easily accumulate in plants [119]. Xie et al. (2014) showed that the rice plant *Oryza sativa* L. is easily contaminated with perchlorate and suggested that perchlorate can inhibit plant growth [120]. Perchlorates also affect the chlorophyll content and root systems of *Acorus calamus*, *Canna indica*, *Thalia dealbata*, and *Eichhornia crassipes* [121]. A study by Acevedo-Barrios et al. (2018) showed that perchlorate significantly reduced the survival of freshwater algae *Pseudokirchneriella subcapitata* (LC50 = 72 mM) [122]. However, the exact way in which perchlorate damages the photosystem is unclear [120].

One of the methods for removing perchlorate from ecosystems is microbial degradation. It is cost effective, easy to implement, and environmentally friendly, making it a viable method for reducing perchlorate pollution. Perchlorate-reducing bacteria (PRB) reduce ClO_4_^−^ or chlorate (ClO_3_^−^) to chlorite (ClO_2_^−^) with perchlorate reductase (*pcrABCD*) and then disproportionate ClO_2_^−^ to Cl^−^ and O_2_ with chlorite dismutase (*cld*) [123] (Figure 3). Electron donors for the reduction of perchlorates are often organic compounds such as methanol and acetate [124,125]. Inorganic donors, such as H_2_ and S, are also capable of causing the reduction of perchlorates [126,127]. Moreover, researchers have recently reported that PRBs are able to reduce perchlorates using methane as an electron donor [128,129,130].

The reduction of perchlorate is usually inhibited by the presence of nitrates [125,131], as some reducing micro-organisms prefer other electron acceptors to perchlorates [132]. To prevent this, donors are added in excess to remove non-perchlorate electron acceptors before reduction is performed; this is carried out because non-perchlorate electron acceptors can activate bacteria that do not degrade perchlorate, resulting in inefficient processing. Oxygen is another inhibitor of microbial perchlorate reduction, as its presence can cause bacteria to use donors to consume oxygen [132,133]. Research showed that perchlorate recovery should ideally be performed under facultative anaerobic conditions [134,135].

*Azospirillum* strains capable of degrading perchlorate have been repeatedly isolated from samples contaminated with perchlorate. At the same time, they could use acetate [136,137] or hydrogen [138] as electron donors. It was recently shown that in a batch membrane biofilm reactor, representatives of the genus *Azospirillum*, along with the genus *Denitratisoma*, were the main genera involved in the reduction of perchlorates and nitrates, and both were able to use NO_3_^−^ and ClO_4_^−^ as electron acceptors [129].

The ability of bacteria from the genus *Azospirillum* to biodegrade perchlorate makes it possible to use them for the remediation of contaminated soils, and therefore, the negative effect of perchlorate on plants can be reduced.

### 3.7. Osmotic Stress

Osmotic stress in a plant cell occurs when the concentration of the solvent (water) in the environment is lower than in the cell. This is possible in two cases: with salinity and with drought. The physical way to reduce osmotic stress is the synthesis of osmolytes—low molecular weight organic substances that are soluble in the intracellular environment and change the properties of biological fluids. The main osmolytes are prolines, soluble sugars, and glycine–betaine [139].

Proline has very strong moisturizing properties. Its hydrophobic part is able to bind to proteins, while its hydrophilic part is able to bind to water molecules, allowing proteins to access more water to increase their solubility and prevent protein denaturation through dehydration under osmotic stress conditions [140]. Trehalose is a reducing disaccharide. Under the conditions of drought stress, the intercellular content of trehalose rapidly increases, which blocks the transition of the phospholipid bilayer membrane from the liquid crystal state to the solid state and stabilizes the structure of proteins, nucleic acids, and other biological macromolecules [141]. Betaine is a metabolic intermediate belonging to the water-soluble alkaloid compounds of quaternary ammonium. It helps to stabilize the structures and activity of photosynthesis, including protective enzymes, and also helps to maintain membrane integrity from widespread damage under drought stress conditions [142,143,144,145].

Bacteria from the genus *Azospirillum* are not only capable of mitigating the consequences of osmotic stress for plants, but they themselves have a number of mechanisms of resistance to osmotic stress.

The mechanism of osmoadaptation was investigated in relatively more detail in *A. brasilense*, where glycine–betaine was shown to enhance growth and nitrogen fixation under salt stress conditions [146]. In addition to betaine, proline was shown to be the predominant osmolyte at higher salt concentrations [147]. In response to salt stress, a periplasmically located glycine–betaine-binding protein, a component of the ProU system, is induced, which is expressed as one of the “early genes” in the process of osmoadaptation. This protein binds glycine–betaine with a high degree of activity and contributes to its high intracellular accumulation [148,149]. However, Chowdhury et al., 2007, showed that the production of exopolysaccharides and cell aggregates is a more consistent physiological response of *A. brasilense* to salt stress than osmoprotection through glycine–betaine [150]. Nagarajan et al., 2007, also showed that most of the genes induced by salt stress in *A. brasilense* seem to be involved in functions associated with the cell membrane [151].

#### 3.7.1. Drought

Drought stress is one of the major constraints on global agricultural production. Approximately one third of the Earth’s land area is in arid and semi-arid regions, while most of the other land areas are often subject to periodic and unexpected climatic droughts. Water deficit can be fatal to plants and lead to huge social problems and economic losses [152].

Drought stress results in reduced nutrient diffusion, induces the formation of free radicals that affect antioxidant protection, leads to a decrease in chlorophyll content, and affects nitrate reductase activity due to the lower uptake of nitrates from the soil [153]. Drought also enhances ET biosynthesis, which inhibits plant growth [154].

PGPR were shown to reduce the negative effects of drought on plants. It may be due to several factors: the production of phytohormones, such as ABA, gibberellic acid, cytokinins, and IAA; the ability of PGP bacteria with ACC deaminase enzyme to degrade plant ET precursor ACC, thereby reducing ET levels in stressed plants; the induction of systemic tolerance by bacterial compounds though microbe-induced physical and chemical changes in plants that lead to increased resistance to abiotic stresses; and the synthetization through bacteria of exopolysaccharides capable of binding Na^+^ ions [155].

Strains of *Azospirillum brasilense* are most often used as inoculants among the representatives of the genus *Azospirillum* in studies on the negative drought effects on plants (Figure 4). Sometimes, they are used in combination with other PGPRs, mycorrhizal fungi, or zinc or silicon oxide nanoparticles. Studies have also been conducted on *A. baldaniorum* Sp245 [156] (previously *A. brasilense*) and *A. lipoferum* [157].

Azospirilla are able to increase plant resistance to drought stress through the production of auxins [37,158] or through the synthesis of nitric oxide, which acts as a signaling molecule in the IAA-inducing pathway [159,160]. Auxins, in small concentrations, enhance root growth and stimulate the formation of lateral roots. Thus, their effect on the plant leads to an increase in the area of the root system, and therefore, has a positive effect on water absorption and prevents the occurrence of drought stress [158].

ABA is considered to be one of the most important growth regulators involved in osmotic stress signaling and tolerance [161]. ABA accumulates to high levels during drought stress [162]. Data on the effect of azospirilla on the level of ABA in plants are contradictory. On the one hand, *A. lipoferum* has been shown to reduce drought stress through the production of ABA and gibberellins [163]. The level of ABA also increased in Arabidopsis plants inoculated with *A. brasilense* Sp245 [164]. The production of this hormone by *A. lipoferum* increased the concentration of ABA in inoculated maize seedlings (*Z. mays*), which led to stomatal closure [38]. However, stomatal closure inhibits photosynthesis, which leads to the inhibition of plant growth [165].

On the other hand, the inoculation of maize with the *A. brasilense* strain SP-7 in combination with the *Herbaspirillum seropedicae* strain Z-152 under drought conditions led to a decrease in the expression of the ZmVP14 gene, which is involved in the biosynthesis of ABA, and a decrease in the level of ABA in the plant. Additionally, in this work, inoculation caused a decrease in ET levels in corn [39].

One of the ways to stimulate drought resistance in plants through bacteria is to change the elasticity of root cell membranes [158]. It has been shown that *A. brasilense* reduces the membrane potentials of wheat seedlings and the content of phospholipids in cowpea cell membranes due to altered proton efflux activity [166]. Inoculation with azospirilla can prevent an increase in the level of phosphatidylcholine and a decrease in the level of phosphatidylethanolamine in water-deficient conditions in wheat seedlings [167].

Trehalose [168] and the polyamine cadaverine [169] can be mentioned as signaling molecules secreted by azospirilla that stimulate drought resistance in plants. Maize inoculation with *A. brasilense*, which overexpresses the trehalose biosynthesis gene, conferred drought tolerance on maize and significantly increased plant biomass. A very small amount of trehalose is thought to move into maize roots and signal pathways for plant stress tolerance [168].

Another indicator of a decrease in osmotic stress during drought, namely a decrease in the amount of proline, was observed when plants were inoculated with bacteria from the genus *Azospirillum* [38,39,170]. The inoculation of maize plants with *A. lipoferum* improved plant growth by accumulating free amino acids and soluble sugars compared to untreated plants under drought stress conditions [40].

It was also shown that under drought conditions, inoculation with bacteria from the genus *Azospirillum* leads to the activation in plants of enzymatic [157,171,172,173] and non-enzymatic [172,174] antioxidant pathways. Bacterial inoculation also led to lower levels of hydrogen peroxide and lipid peroxidation in plants [170].

The role of polysaccharides in plant adaptation to drought was also shown for members of the genus *Azospirillum*. *A. brasilense* Sp245 capsule material contains high molecular weight carbohydrate complexes (the lipopolysaccharide–protein complex and the polysaccharide–lipid complex) responsible for protection under extreme conditions, such as desiccation. The addition of these complexes to a suspension of decapsulated *A. brasilense* Sp245 cells significantly increased survival under drought stress conditions [175].

So, bacteria from the genus *Azospirillum* are actively used to mitigate the effects of drought in plants. The mechanism of stress factor mitigation is associated with the modulation of the level of phytohormones: auxins, ABA, ET, changes in the elasticity of root cell membranes, changes in the content of osmolytes, the activation of the antioxidant defense system, and the synthesis of polysaccharides.

#### 3.7.2. Salinization

Salinity affects more than 6% of the world’s total land area (approximately 800 million hectares of land worldwide) [176]. Soil salinity has increased due to inefficient irrigation, improper fertilizer application, and industrial pollution [177]. Salinity causes Na^+^ toxicity and ionic imbalance and disrupts vital metabolic processes in plant cells, such as protein synthesis, enzymatic reactions, and ribosome functions [178].

PGPR can mitigate salinity-induced stress in plants through many synergistic mechanisms, including osmotic regulation, the stimulation of osmolyte accumulation and phytohormone signaling, the increase in nutrient uptake, the achievement of ion homeostasis, the reduction of oxidative stress via enhancing antioxidant activity [179], the increased synthesis of volatile organic compounds [180], and improved photosynthesis [76].

Representatives of the genus *Azospirillum* have repeatedly shown their effectiveness in reducing salt stress in plants. The possibility of their use as inoculants under salinity is due to the halotolerance of some strains [181,182]. 

They are used for the inoculation of plants under saline conditions, both alone and in combination with fungi [183], other PGPBs [41,42,184], and even with phosphogypsum [185]. 

The softening effect of inoculation can be manifested in the modulation of the concentration of osmolytes in plants. For example, one of the responses of corn to salinity is the accumulation of a powerful osmolite, i.e., raffinose, in the leaves. The inoculation of plants with *A. brasilense* (HM053) resulted in a decrease in the content of raffinose in the leaves and an increase in the content of sucrose [186]. Inoculation with azospirilla also improves the content of soluble sugars and proline in plants [42,44] and increases the content of glycine–betaine [187] under salt stress.

In addition, azospirilla can increase the K^+^/Na^+^ ratio in plants [41,42,43,183,184,188]; increase the content of nitrogen, phosphorus, calcium [183,188], and magnesium [183] in the crop; increase the content of nitrates; and reduce the content of chlorides [184], as well as increase the activity of nitrogenase and phosphatase [183] under salinity.

Additionally, inoculation with azospirilla leads to an improvement in the morphological characteristics of plants [41,42,43,44] and an increase in yield [41,185] and protein content [42,183] under saline conditions.

Azospirilla also affects the level of oxidative stress in plants under saline conditions. This results in a decrease in the content of malonic aldehyde [42,43] and hydrogen peroxide [42]. Data on the effect of azospirilla on antioxidant defense enzymes under salt stress are contradictory. They can cause both an increase [41,44,185,187] and a decrease [41,42] in the activity of antioxidant enzymes.

In most studies, an increase in the content of chlorophylls and carotenoids was shown when plants were inoculated with azospirilla under salt stress conditions [41,42,185]. However, Del Amor and Cuadra-Crespo, 2012, showed that the co-inoculation of *A. brasilense* and *Pantoea diversa* on sweet peppers did not affect the photochemical efficiency of photosystem II and the relative content of chlorophyll but contributed to maintaining a higher stomatal conductivity; therefore, they concluded that the influence of inoculants on the response to salinity was due mainly to the stomatal regulation of photosynthesis and not to the influence on the biochemical limitations of photosynthesis [184].

It was also shown that the most important compounds of secondary metabolism (phenylpropanoids, alkaloids, and other N-containing metabolites, as well as membrane lipids) and phytohormones (brassinosteroids, cytokinins, and methyl salicylate) showed the most pronounced modulation in response to treatment with azospirilla under salt stress [44]. Thus, the effect that azospirilla inoculation has on plants can be varied, but in most cases, it leads to the mitigation of salt stress.

## 4. Conclusions

Understanding the mechanisms of the positive effects of bacteria from the genus *Azospirillum* on plants under conditions of biotic and abiotic stress is of great importance due to the increasingly active use of this bacterial group as bioinoculants of agricultural plants. In recent years, more and more new *Azospirillum* strains and species have been described, each of which has the potential to be an interesting biopreparation for mitigating different types of stress in plants. Different groups of authors evaluated different plant parameters during inoculation with azospirilla under stress conditions. In general, the mitigation of biotic stress was carried out using azospirilla through the synthesis of siderophores and the induction of systemic resistance in plants, and mitigation of the effects of abiotic stress was carried out through the modulation of the level of phytohormones, osmolytes, and volatile organic compounds in the plant and in regard to the efficiency of photosynthesis and the antioxidant defense system. Increasing the resistance of pests and phytopathogens to agrochemicals, as well as to global warming, which is leading to higher temperatures and increased dry periods, results in the need to use stress-resistant inoculants. In the future, it seems possible to test the ability of the *Azospirillum* species described in recent years to reduce the impact of stress factors on plants and to test the use of *Azospirillum* in combination with other micro-organisms. This review of the data obtained to date will allow researchers to facilitate the design of new experiments and accelerate the implementation of results in practice.

## Figures and Tables

**Figure 1 ijms-24-09122-f001:**
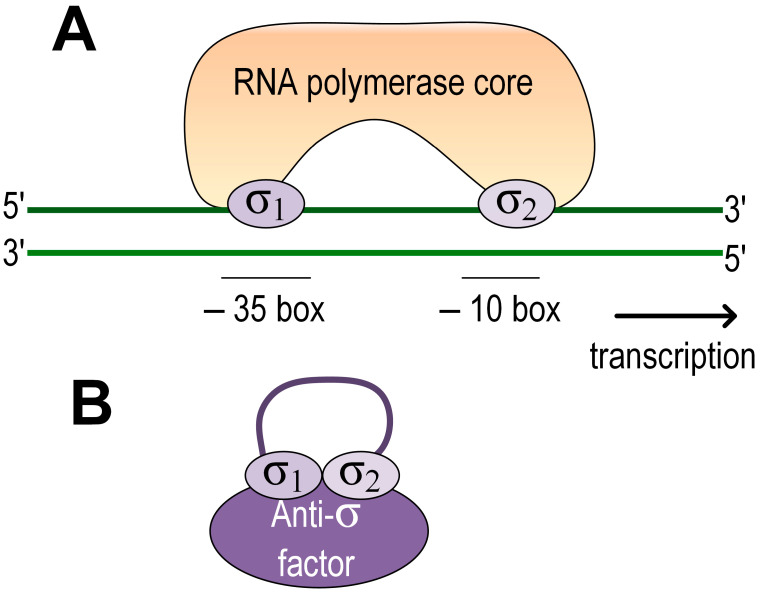
Scheme of ECF regulation in *Azospirillum*. (**A**) Inactive state. ECF is associated with the anti-sigma factor. (**B**) Active state. The σ 2 domain binds to the promoter at the −10 box and the σ4 domain at the −35 box. DNA begins to melt from the −10 to the start codon.

**Figure 2 ijms-24-09122-f002:**
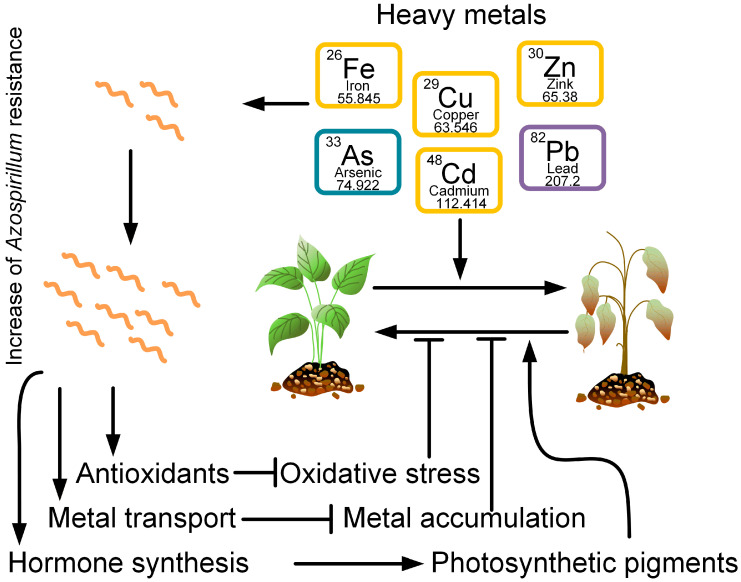
Heavy metals, such as iron, arsenic, copper, cadmium, zinc, and lead, have negative effects on plant growth and vitality. However, at the same time, certain *Azospirillum* strains can be resistant to these metals. *Azospirillum* produces antioxidants that neutralize oxidative stress induced by heavy metals. *Azospirillum* also promotes the transport of heavy metals from plant cells, preventing its intracellular accumulation. In addition, *Azospirillum* produces plant hormones that promote the formation of photosynthetic pigments. Together, the reduction of oxidative stress, the removal of metals from plants, and the synthesis of photosynthetic pigments contribute to an increase in plant resistance to heavy metal stress.

**Figure 3 ijms-24-09122-f003:**
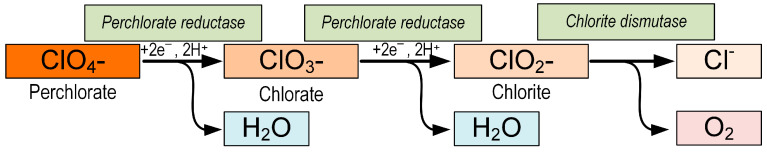
Reactions of perchlorate reduction by perchlorate-reducing bacteria. Perchlorate reductase catalyzes perchlorate reduction to chlorate and chlorate reduction to chlorite. Chlorite dismutase catalyzes chlorite dismutation to chloride and oxygen.

**Figure 4 ijms-24-09122-f004:**
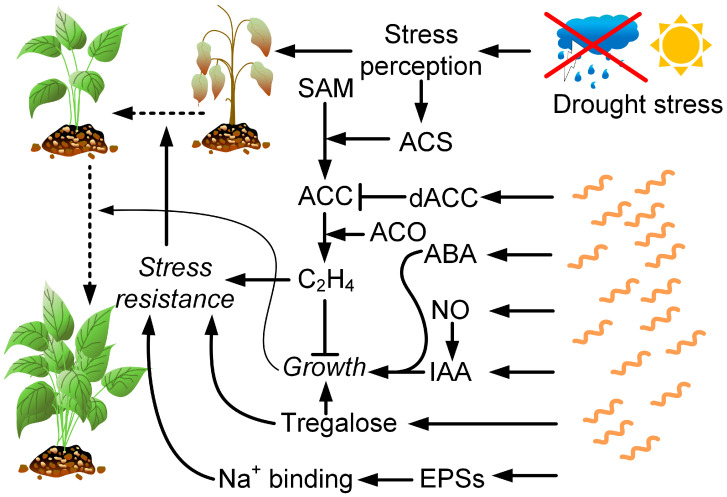
Drought has a complex negative effect on plants. However, some defense mechanisms are activated. For example, drought activates ACS (ACC synthase), which catalyzes the formation of ACC (1-aminocyclopropane-1-carboxylic acid) from SAM (S-adenosyl-L-methionine). Further, ethylene (C_2_H_4_) is formed from ACC by ACO—ACC oxidase. Ethylene, through a variety of mechanisms, increases plant resistance to drought, but at the same time, limits their growth, which can adversely affect crop productivity. The bacteria of the genus *Azospirillum* produce ACC deaminase (dACC), thereby limiting ethylene synthesis in plants. In addition, they produce ABA and IAA, as well as nitric oxide, which contributes to the synthesis of IAA. Together, these factors cause the induction of plant growth even in drought conditions. *Azospirillum* produces trehalose, which simultaneously promotes plant growth and increases its resistance to drought. *Azospirillum* also synthesizes exopolysaccharides capable of binding Na^+^ ions.

**Table 1 ijms-24-09122-t001:** Influence of inoculation with *Azospirillum* strains on the yield of agricultural plants under stress.

The Crop	Stress Type	*Azospirillum* Species	The Percentage of Improved Growth or Yield	Reference
wheat	arsenic	*A. brasilense*	plant height 2.36–3.21% spike length 11.42–22.19% number of spikelets per spike 4.46–6.60% number of grains per spike 4.67–5.69% 1000 grain weight 5.17–9.63% grain yield per plant 3.42–17.6%	[25]
arabidopsis	cadmium	*A. brasilense*	shoot fresh weight about 100%	[26]
pak choi	cadmium	*A. brasilense*	biomass 26–255%	[27]
barley	cadmium	*A. lipoferum*	root biomass 22.22% root elongation 12.5%	[28]
pak choi	cadmium	*A. brasilense*	shoot biomass 16.2% root biomass 12.2%	[29]
cucumber	copper	*A. brasilense*	root weight 55.32% root length 73.65% root tips 35.85%	[30]
tomato	*Pseudomonas syringae* pv. *tomato*, the causal agent of bacterial speck on tomato	*A. brasilense*	dry weight about 100%	[31]
tomato	*Pseudomonas syringae* pv. *tomato*, the causal agent of bacterial speck on tomato	*A. brasilense*	dry weight 7.81–28.79%	[32]
green gram	nematode disease	*A. lipoferum*	shoot length 10.26% fresh weight 18.28% dry weight 18.45%	[33]
cherry tomato	*Clavibacter michiganensis* subsp. *michiganensis* (bacterial canker), *Xanthomonas campestris* pv. *vesicatoria* (bacterial spot)	*A. brasilense* and *Azospirillum* sp. BNM-65	leaves 32–43% shoot height 12–143% shoot dry weight 81–107% root dry weight 37–80%	[34]
teosinte	fungal diseases caused by *Alternaria*, *Bipolaris* and *Fusarium*	*A. brasilense*	total dry mass from −8.6 to 73.0%	[35]
komatsuna	radioactive 137Cs	*Azospirillum* sp. strain TS13	dry weight 40–51%	[36]
wheat	drought	*A. lipoferum*	wheat yield up to 109%	[37]
arabidopsis	drought	*A. brasilense*	rosettes diameter 7.7% rosettes DW 86.21% seed yield 328.66%	[38]
maize	drought	*A. brasilense*	total biomass 26%	[39]
maize	drought	*A. lipoferum*	height 35.33–43.89%	[40]
coriander	salinization	*A. brasilense* and *Azotobacter chroococcum*	grain yield 11.6% stem fresh weight 11.3% stem dry weight 17.2% total plant fresh weight 6.1% total plant dry weight 10.2%	[41]
flax	salinization	*A. brasilense*	shoot length 16.5% root length 36.6% fresh weight of shoot 17.07% dry weight of shoot 13.43% fresh weight of root 57.7% dry weight of root 78.6% number of leaves 10.5%	[42]
white clover	salinization	*A. brasilense*	shoot height 57.8–70% root length 58.82–70.85%	[43]
tomato	salinization	*A. brasilense*	root biomass 118%	[44]

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
