# Peer review of "Molecular Mechanisms Determining the Role of Bacteria from the Genus Azospirillum in Plant Adaptation to Damaging Environmental Factors"

_ijms, 2023, doi:10.3390/ijms24119122_

Round 1

Reviewer 1 Report

Journal:              IJMS (ISSN 1422-0067)

Manuscript ID:   ijms-2399152

Type:                 Review

Title:                  Molecular mechanisms determining the role of bacteria from the genus Azospirillum in plant adaptation to damaging environmental factors

Authors:   Maria V. Gureeva , Artem P. Gureev *

Section:               Molecular Microbiology

Special Issue:     Latest Reviews in Molecular Toxicology 2023

Comments

The Review article is well presented.  However, there are a few queries,

Q1: In addition to Azospirillum spp., there are quite a few organisms that have PGPR characteristics. Hence, before focusing on one organism, we need to evaluate and compare it to others. Why has it been selected for this review?

Q2: Table 1 has no significance. It may be deleted.

Q3: Line 126 -129

Wu et al., 2021 isolated two Azosp …………………….. [34]

Young et 127 al., 2008 isolated from  …………………….[28]

Eckford et al., 2002 isolated A. brasilense among……………. [51].  

Al-Mailem et al., 2019, found ….. [58]

Saeed 135 et al., 2021 used A. oryzae (MS6) as part  …………….. [59]

This style of writing a review is not impressive. There are many such cases throughout the Review article. The information needs to be compiled and presented as a paragraph with References, where ever required.

Q4: Lines 293- onwards.  3.4. Pesticides pollution

      Line 304: Data on pesticide toxicity for azospirilla are inconsistent.  

      References 114:  1996.  115: 2004.  116: 2000   pertain to Azospirillum but are quite old.

     References 120:  Aerobic and Anaerobic cultures

     Ref.121: Multiple bacterial cultures: “……Pseudomonas, Sordaria, Caulobacter, Magnetospirillum, Rhodospirillum were abundant. While, genera Actinoplanes, Streptomyces, Bradyrhizobium, Rhizobium, Azospirillum, Agrobacterium, ….”

     Ref.122: MFC – Multiple bacterial cultures “…..  The bacteria possessing functions of compounds degradation (e.g. Petrimonas, Desulfovibrio, and Mycobacterium) and electrons transfer (e.g. Petrimonas, Cloacibacillus, and Azospirillum) w   …..”

       Ref 123: Multiple bacterial cultures:  “…..microbial community analysis revealed that the functional bacteria Sphaerochaeta, Pseudomonas, Azospirillum, Azoarcus, and Chryseobacterium were major predominant bacteria in the anodic biofilm.    “

These References do not strongly support the role of  Azospirillum in Pesticides pollution

Q5: Line 374 onwards   3.7. Osmotic stress

Most of the References cited in this section do not strongly support the role of  Azospirillum

The presentation is oriented towards supporting Azospirillum.

English is OK

Reviewer 2 Report

I have thoroughly read the manuscript. The authors have addressed a significant topic and provided comprehensive coverage of it. However, this review does have some limitations, which are mentioned in the attached annotated file.

Kindly revise the manuscript carefully, as I recommend a major revision.

I would like to review the revised version to ensure that the suggested changes have been incorporated. 

Kindly check the grammar and typos.

Please use short sentences.

Ensure that these requests are followed throughout the manuscript.

Round 2

Reviewer 1 Report

Accept

Reasonably good

Reviewer 2 Report

The authors have thoroughly revised the manuscript. Now, the paper can be published in its present form.